# Sustainalism: An Integrated Socio-Economic-Environmental Model to Address Sustainable Development and Sustainability

**N. P. Hariram** [1], **K. B. Mekha** [2], **Vipinraj Suganthan** [3] and **K. Sudhakar** [4,5,6,*]

1. Renewable Energy and Environmental Engg Focus Group, Universiti Malaysia Pahang, Paya Basar 26300, Malaysia
2. Integrated Centre for Green Development and Sustainability (ICFGS), Cuddalore 607001, India
3. Faculty of Engineering and Natural Sciences, Tampere University, P.O. Box 541, 33014 Tampere, Finland; vipinrajsugathan@tuni.fi
4. Faculty of Mechanical and Automotive Engineering Technology, Universiti Malaysia Pahang, Pekan 26600, Malaysia
5. Centre for Research in Advanced Fluid & Processes (Fluid Centre), Universiti Malaysia Pahang, Paya Basar 26300, Malaysia
6. Energy Centre, Maulana Azad National Institute of Technology, Bhopal 462003, India
* Correspondence: sudhakar@ump.edu.my

**Abstract:** This paper delves into the multifaceted concept of sustainability, covering its evolution, laws, principles, as well as the different domains and challenges related to achieving it in the modern world. Although capitalism, socialism, and communism have been utilized throughout history, their strengths and drawbacks have failed to address sustainable development comprehensively. Therefore, a holistic approach is necessary, which forms the basis for a new development model called sustainalism. This study proposes a new socio-economic theory of sustainalism that prioritizes quality of life, social equity, culture, world peace, social justice, and well-being. This paper outlines the six principles of sustainalism and identifies sustainalists as individuals who embrace these new concepts. This study also explores how to attain sustainalism in the modern world through a sustainable revolution, representing a step toward a sustainable era. In conclusion, this paper summarizes the key points and emphasizes the need for a new approach to sustainalism in the broader sense. The insights provided are valuable for further research on sustainalism and sustainability.

**Keywords:** sustainalist; sustainability; sustainable revolution; SDG; quality of life; sustainalism

## 1. Introduction

*Evolution of Sustainable Development and Sustainability*

Sustainability has been widely accepted since the olden days, especially in rural societies. The world's ancient cultures combine worship and religious convictions with environmental preservation, which calls on people to take care of the planet and keep it in good condition; this may be considered a demonstration of sustainability in the ancient ages. The "sustainability" term's origins can be found in the realm of hunting, wherein hunters and gatherers were eager to establish a stable means of subsistence. In old German, "sustenance" refers to provisions kept in reserve for emergencies. The verb "to sustain" or the phrase "sustainable" have both been "proven to be a derivation of the noun "sustenance" (literally retain, what one retains). Nowadays, the word "sustainable" still has the meaning of being "enduringly effective" in common usage [1].

*Silent Spring* by Racheal Carson, The Ecologist's *A Blueprint for Survival,* and *The Population Bomb* by Ehrlich are some early works that significantly impacted the world in the cradle stage of sustainable development during the 1960s. After that, within a short time, the words "sustainable" and "sustainability" were introduced

in the Oxford English Dictionary [2]. The word "sustainable" comes from the Latin word "sustinere". Thomas Malthu's postulates on the drastic consumption of natural resources and energy emerged, addressing the aftermaths of the population explosion. In his essay, he stated the principle of population, showing that population growth is not sustainable, which is not in proportion with the available resources and carrying capacity of the Earth [3].

The future and existence of humanity were described as "sustainability" in the British book *Blueprint for Survival* and a United Nations statement in 1978. Policy journals began using the word "sustainability" along with technical articles and studies around 1978. Most of these concentrated on the major sustainability domains and the environment. Soon, the World Bank started working to integrate sustainability into its organizational structure, operational procedures, and policy frameworks. Due to the fact that the term "sustainability" has roots in so many fundamentally different ideas, each with a compelling argument for its legitimacy, it seems futile to attempt to define it in a single sense [4]. The concept of sustainable development has acquired acceptance and significance theoretically. Its further development is frequently overlooked or minimized. While some people may think evolution is irrelevant, it may still be used to predict future trends and defects, which can be helpful now and in the future [5]. The unchecked economic expansion may cause the planet's carrying capacity to be exceeded and civilization to crumble. The ideas of sustainability and sustainable development, as a result, emerged [6].

The repercussions of anthropogenic activity and environmental devastation are becoming increasingly well-known, thanks to the media and publications. Works such as *Limits of Growth* or *Small is Beautiful* argued that economy-based development is unsustainable in this finite world of limited resources, and this started to question ongoing economic growth [7]. The early discourse was radical and demanded structural reform, arguing that capitalist economic development cannot be integrated with social and ecological development, which contradicts the concept of a sustainable world [8].

Reiterating the need for SD, the "World Commission on Environment and Development", headed by Gro Harlem Brundtland of Norway, produced the Brundtland Report titled "Our Common Future" in 1987. The report defined sustainable development as "the development that meets the demands of the current generation without compromising the ability of the future generation to meet their own needs", as was already mentioned. The Rio Earth Summit, also known as the UNCED or Rio Earth Summit, was inspired by the Brundtland Report in 1992 [9]. The main subject of discussion at the UNCED was the report's recommendations. The conference outcome document for the UNCED included Agenda 21 as one of the critical sustainable development outcomes. It urged that national policies be devised and implemented to address the economic, social, and environmental components of sustainable development after stating that sustainable development should become an essential item on the international community's agenda [10]. The World Summit on Sustainable Development (WSSD), also known as Rio+10, was convened in Johannesburg in 2002 to assess the status of putting the Rio Earth Summit's outcomes into practice. The World Summit on Sustainable Development (WSSD) introduced several multi-stakeholder partnerships for sustainable development and the Johannesburg Plan, an implementation plan for the measures outlined in Agenda 21 [11]. Figure 1 illustrates the sequence of activities associated with sustainable development.

## Activities Up to SDGs

Environmental Conservation
Action Plan Published.

**UNEP Conference on the Human
Environment**

Developing Measures on
impact of Climate change

**1st World Climate Conference**

WCED; Implement development plans
on environmental conservation

**UN's World Commission on
Environment & Development**

1972

1979

1984

1969

1975

1981

**UN Report : Man and His
Environment**

Activities initiated
to address environmental
deterioration.

**International Congress of the
Human Environment (HESC)**

World Wide environmental
conservation.

**1st UN Conference on Least
Developed Countries**

Guidelines for supporting
under developed countries.

Working guidelines for
sustainability concepts for future

**The World Summit on Sustainable
Development**

Kyoto protocol : Reduce $CO_2$
& green house gases

**Kyoto Climate Change Conference**

2002

1997

1987

2009

2000

1992

**Our Common Future /
Brundtland report**

Introduced Basic
Principles and
Basics of
Sustainability

**World Congress
Summit G20**

G20 Countries focused
on a moderate and
sustainable economy

**UN's Millennium Declaration**

Introduced 8 Millennium
Development Goals (MDGs)

**Earth Summit /
Rio Conference**

Agenda 21 Action Plan
principles of sustainable
development

Paris climate change
conference: limiting global
warming by controlling GHGs.

**UN conference on climate
change, COP21**

2015

2012

2015

**UN conference Rio +20**

Renewed "The future we
want" focused on the goals
of sustainable development

**UN sustainable development
summit 2015**

Published UN 2030 agenda for
sustainable development

**Figure 1.** Overview of the various activities related to the concept of sustainable development till SDGs [12].

Rio+20, also known as the United Nations Conference on Sustainable Development (UNCSD), occurred in 2012, 20 years after the first Rio Earth Summit. The conference's two main sustainable development topics were the green economy and

an institutional framework. The conference conclusion document "The Future We Want" placed a strong emphasis on sustainable development to the point that the term "sustainable development" was used [13]. The Rio+20 outcomes included a procedure for creating new SDGs, which would go into effect in 2015 and promote targeted action regarding sustainable development in all areas of the global development agenda. SD was thus one of the five main objectives of the United Nations in 2012 (UN), highlighting the important part that sustainable development (SD) should play in national and international development policies, programs, and agendas [14]. Table 1 provides an overview of the different definitions of sustainability and sustainable development.

**Table 1.** Different definitions of sustainability and sustainable development [15–18].

| Source | Definition of Sustainability |
|---|---|
| The Brundtland Report | A process of change in which the exploitation of resources, the direction of investments, the orientation of technological development, and institutional changes are made consistent with future as well as present needs. |
| Earth Centre | Sustainability means that all living things on Earth have obligations to each other, the larger biosphere, and the subsequent generations. |
| NCARB | In sustainability, interrelated ecological, economic, and social systems succeed now without sacrificing their future prosperity. |
| UN | Sustainability is meeting the demands of the present without compromising the ability of future generations to satisfy their own needs. |
| Hannover principles | Sustainability is the conception and realization of ecologically, economically, and ethically sensitive as well as responsible expression as a part of the evolving matrix of nature. |
| Source | Definition of Sustainable Development |
| WCED | Sustainable development is the development or growth which meets the needs of the present without compromising the ability of future generations to meet their own needs. |
| Berke and Manta | Sustainable development is defined as a dynamic process connecting local and global concerns, as well as linking local social, economic, and ecological issues, to cater to the current and future generations' needs fairly. |

## 2. Laws and Principles of Sustainability

The term "sustainability" is scrutinized by Albert Bartlett using different laws, hypotheses, observations, and predictions. They may not apply to small groups of people or to tribes living in primitive conditions as they are all based on populations and rates of resource and good consumption found in the world. The laws are more exacting than the hypotheses [19]. Though they are limited in many perspectives, some postulates may be proven correct by experience and given the status of laws. The observations may provide insight into the issues and possible solutions. These postulates can be classified into four categories: Population and consumption, Energy, Resource and Environment, and Human- and Society-centric (Table 2).

**Table 2.** Albert's laws on sustainability [20].

| | |
|---|---|
| Population and Consumption | First Law: Population growth and increase in the rates of consumption of resources cannot be sustained. |
| | Second Law: The difficulty of transforming a society, with more growth in population and higher consumption of resources, into being sustainable is higher. |
| | Third Law: Population Momentum: The response time of populations to changes in the human fertility rate is the average length of a human life. |
| | Fourth Law: The size of the population that can be sustained (the carrying capacity) and the sustainable average standard of living of the people are inversely related. |
| | Eighth Law: Sustainability requires that the size of the population be less than or equal to the carrying capacity of the ecosystem for the desired standard of living. |
| | Ninth Law: The benefits of population growth and growth in the rates of consumption of resources accrue to a few; all of society bears the costs of population growth and growth in the consumption of resources. |
| | Seventeenth Law: If, for whatever reason, humans fail to stop population growth and growth in the rates of consumption of resources, nature will eliminate these growths. |
| Energy, Resource, and Environment | Tenth Law: Growth in the rate of consumption of a non-renewable resource, such as a fossil fuel, causes a dramatic decrease in the life expectancy of the resource. |
| | Eleventh Law: The time of expiration of non-renewable resources can be postponed, possibly for a very long time. |
| | Twelfth Law: When considerable efforts are made to improve the efficiency with which resources are used, the resulting savings are wholly and rapidly wiped out by the added resources consumed due to modest population increases. |
| | Thirteenth Law: The benefits of large efforts to preserve the environment are rapidly canceled by the added environmental demands resulting from small increases in the human population. |
| | Fourteenth Law: (Second Law of Thermodynamics) When rates of pollution exceed the natural cleansing capacity of the environment, it is easier to pollute than it is to clean up the environment. |
| Human—an Society-centric | Seventh Law: A society that has to import people to do daily work ("We can't find locals who will do the work") is not sustainable. |
| | Sixteenth Law: Humans will always be dependent on agriculture (This is the first of Malthus' two postulates). |
| | Eighteenth Law: In local situations within the State, creating jobs increases the number of people locally who are out of work. |
| | Nineteenth Law: Starving people do not care about sustainability. |
| Universal | Fifth Law: One cannot sustain a world in which some regions have high standards of living while others have low standards of living. |
| | Sixth Law: All countries cannot simultaneously be net importers of carrying capacity. |
| | Fifteenth Law: (Eric Sevareid's Law): solutions are the chief cause of problems. (Sevareid 1970) |
| | Twentieth Law: The addition of the word "sustainable" to our vocabulary, to our reports, programs, and papers, to the names of our academic institutes and research programs, and to our community initiatives is not sufficient to ensure that our society becomes sustainable. |
| | Twenty-first Law: Extinction is forever. |

*Principles of Sustainability*

Sustainable development can only be realized if a few principles are followed. However, the economy, environment, and society are typically prioritized when discussing the basics of sustainable development [21]. Population control, human resource manage-

ment, ecological and biodiversity preservation, production systems, the preservation of progressive culture, and public participation are among the issues addressed [22].

One of the tenets of sustainable development is the preservation of the environment. Since all life would end without the environment and biodiversity, they must be protected. The Earth's finite resources cannot meet the population's needs and means. Natural resource extraction must not exceed the Earth's capacity for sustainable development because resource depletion hurts the ecosystem [23]. This suggests that development activities need to consider the Earth's capabilities. Due to this, having renewable energy sources, such as solar, is essential, rather than relying too heavily on hydroelectricity and things made from petroleum. To accomplish sustainable development, population control is also crucial [24]. People can live by using the limited resources of the Earth. The growing population raises human needs, such as those for food, clothing, and housing, but there are limits to how much the world's resources can be expanded to provide. Therefore, population management and control are essential for sustainable development [25]. Effective human resource management is another integral component of sustainable development. The individuals are in charge of seeing that the principles are upheld. The environment must be used wisely and protected by humans. It is up to individuals to maintain peace on Mother Earth [13]. The argument is built on the premise that sustainable development cannot be accomplished solely through the efforts of one person or organization, which alludes to the system's theory. All individuals and relevant organizations must share this obligation. The concept of participation, which calls for optimistic attitudes from the populace in order to make real progress while accepting responsibility and accountability for stability, is the cornerstone of sustainable development [24]. In order to achieve genuine, long-lasting change, participation entails the combined effort of numerous people and organizations who are all working toward a shared vision of sustainability. We are more likely to succeed if we recognize the responsibility that is placed on each of us as well as the power that comes from working together for a common goal [26].

Sustainable development requires promoting socially progressive traditions, behaviors, and political cultures. In order to maintain social cohesion and support environmental appreciation and preservation for sustainable development, advanced traditional and political culture must be developed, nurtured, and expanded [22]. The systematic integration of ecological, social, and economic considerations into all areas of outcome across generations can be summed up as the central tenet of sustainable development. A socially progressive culture is crucial for sustainable development because it enables people to understand their obligations to society and the environment. Therefore, in order to achieve sustainable development, a progressive traditional and political culture must be created. This can be achieved by implementing measures such as encouraging grassroots organizations to increase public awareness and participation in sustainable development initiatives [27].

A systematic consensual and heuristic approach was used to arrive at sustainability principles based on studies of humans as a social species, the Laws of Thermodynamics, and the science behind them [28]. The lack of a comprehensive definition of "sustainability" and the recognition of the inherent issues involved in the current use of the term "sustainable development" served as the inspiration for this [29]. These pillars support several logical deductions about how social and ecological systems communicate. The guiding principles gradually evolved and were consented to after discussions with experts from the larger scientific community. A framework with logical guiding principles was used to apply the system conditions (Figure 2). These concepts effectively outline how the system parameters may be addressed using "back-casting".

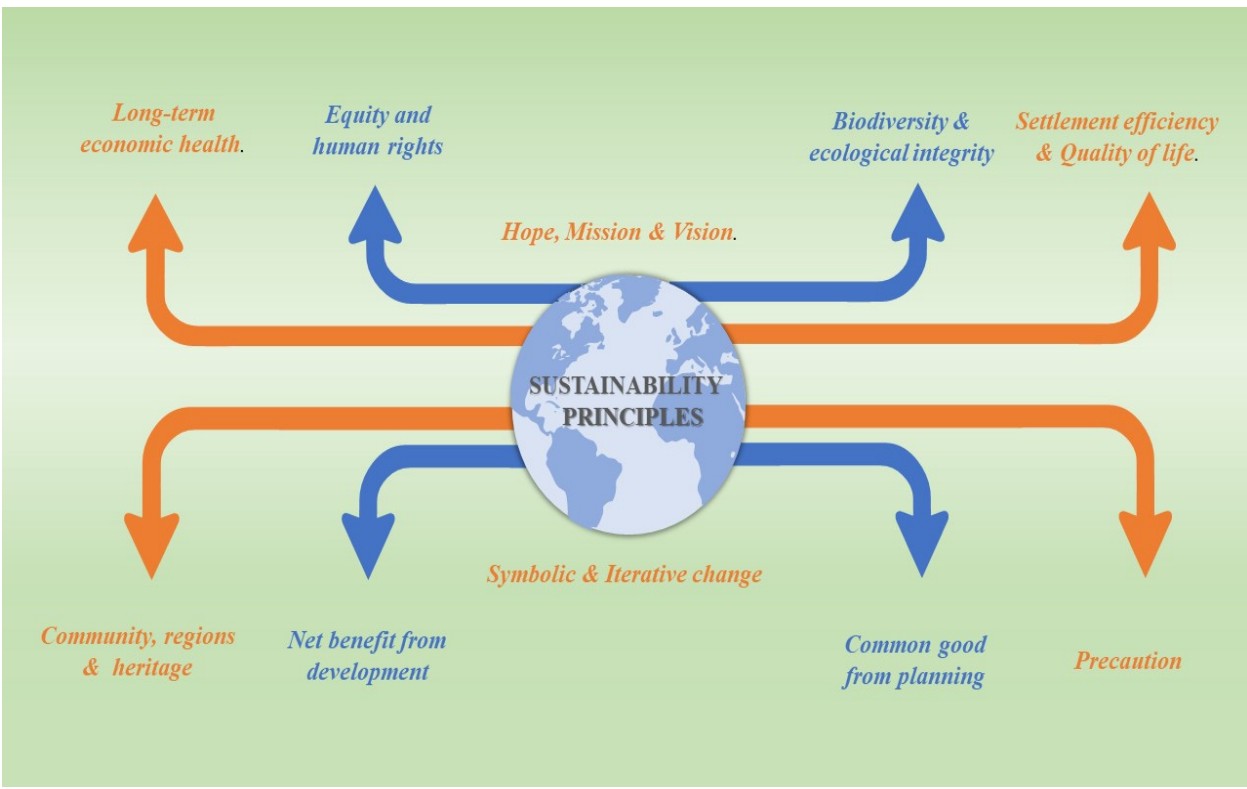

**Figure 2.** Sustainability principles.

In essence, they restate the definition of sustainability in a format that is applicable and relevant to all people, reclaiming it from the hazy "definition drift" observed for the term "sustainable development". As a result, they act as unyielding "custodians" of the sustainability idea [30].

## 3. Three Pillars and Domains of Sustainability

A sustainable structure is said to be built on three pillars. Three intersecting circles are a common visual depiction of sustainability and its dimensions [31]. Many research findings also use a nested approach with a particular dimension at the center. Present-day sustainable development is frequently represented by literal "pillars" supporting it. The hierarchy of the dimensions is highlighted in the schematic with the nested ellipses, with "environment" serving as the basis for the other two. Three interconnected "pillars", "dimensions", "components", "stool legs", "aspects", "perspectives", etc., are frequently used to describe sustainability and include economic, social, and environmental (or ecological) factors or "goals" [32]. It must be acknowledged that these conflicting terms are typically used synonymously, and our preference for "pillars" is largely arbitrary. The three intersecting circles of society, environment, and economy are frequently, though not always, used to represent this multi-stakeholder description, with sustainability situated at the crossroads. While frequently referred to as a "Venn diagram," this diagram frequently lacks the particularly emphasized attributes associated with such a construction. It describes "sustainability" in academic literature, policy documentation, business literature, and online [33].

Alternative ways of expressing the three concepts include using nested concentric circles or actual "pillars" to represent them visually and using them independently of visual aids to represent distinct categories of sustainability objectives or metrics [34]. While captivating due to their simplicity, the meaning these diagrams and the larger "pillar" concept themselves convey is frequently ambiguous, restricting their ability to be coherently operationalized. However, the conceptual underpinnings of this description and the time when it entered popular culture are unclear, and its precise meaning is up for debate. The

three-pillar conception has undoubtedly attained widespread acceptance, but this should not obscure its flaws [7].

Critics argue that the sustainable development framework is not ambitious enough to address the scale and urgency of today's environmental and social challenges. Some also argue that focusing on economic growth and development can perpetuate the unsustainable use of natural resources and the unequal distribution of wealth and power. Some alternatives to traditional sustainable development have been proposed, including "Degrowth" and "Post-Development" [35,36]. Degrowth advocates for a reduction in consumption and production, particularly in wealthy countries, to reduce pressure on the environment and address social inequalities. Post-Development argues that the focus on economic growth has created a distorted view of development that prioritizes Western values and disregards the knowledge and values of marginalized communities [35,36]. Ultimately, the debate around sustainable development and its alternatives highlights the need for a more nuanced and inclusive approach to development that prioritizes both human well-being and environmental sustainability. In order to better navigate the turbulent and uncertain conditions that make up the post-Brundtland world, academics, development practitioners, environmental managers, sustainability advocates, and government planners must work together [37].

Since the Brundtland Report was released, mainstream sustainable development has advanced rapidly. The notion of sustainable development is firmly rooted in many government offices, corporate boardrooms, and the hallways of international NGOs and financial institutions, despite the risk of cooptation and abuse, frequently resulting in a scaling-back of its more progressive prescriptions for achieving sustainability [38]. At the very least, its willingness to offer some commonality for deliberations among various development and environmental sectors, which are frequently at odds, can be used to explain why sustainable development has endured. Strongest proponents of the idea, such as those in international environmental NGOs and intergovernmental organizations, are thus at ease advancing a concept that most effectively converts former opponents into social constructivism, contending that understanding the world invariably entails a series of mediations between human social relations and individual identities. Critics also tend to conduct qualitative research based on a case study methodology and emphasize the historical contingency of development processes. Perhaps most significantly, proponents of traditional sustainable development still view the policy-making process as a legitimate means of reform [39].

### 3.1. Domains of Sustainability

The framework assumes that three domains—the economy, the environment, and the social domain—should be considered when discussing sustainability [34]. These domains are claimed to connect as three separate spheres of life. Sustainability is foundational to public administration, policymaking, and political governance; variations of this three-domain framework can be found throughout all policy documents [40]. Despite this prevalence, the framework is infrequently subjected to sustained or in-depth stakeholder discussion. The economy (or profit), the environment (or ecology, the planet), and the social sector (or society, prosperity) are typically left as the framework for defining and operationalizing sustainability, with the economy typically leading the pack as the first between many "equals". This approach to sustainability is often seen as one that places too much emphasis on economic prosperity while not giving enough attention to social and environmental aspects. The idea of sustainability has become ubiquitous in the public sector, but its implementation and application often fail to capture the complex realities of social and environmental interdependencies [41]. In other words, economic factors have taken over almost all decision-making processes. They are viewed as fundamental to the human condition, defining and serving as a standard by which everything else is measured [42].

The three domains are considered separate actual activity spheres. Despite the justifications for why the three domains must be combined in an integrated assessment, they are still hypothesized as different spheres, pillars, or circles (Figure 3). This does not mean that various sustainability domains should not have their own integrity and measurement methods. However, in order to have it both ways, it is necessary to name different sustainability domains analytically and acknowledge that, in reality, they are dimensions of a whole rather than separate spheres that ought to be reconnected [43]. To do this, an integrated approach to sustainability needs to be adopted in which these different domains are conceptualized as parts of a system, not stand-alone spheres [44].

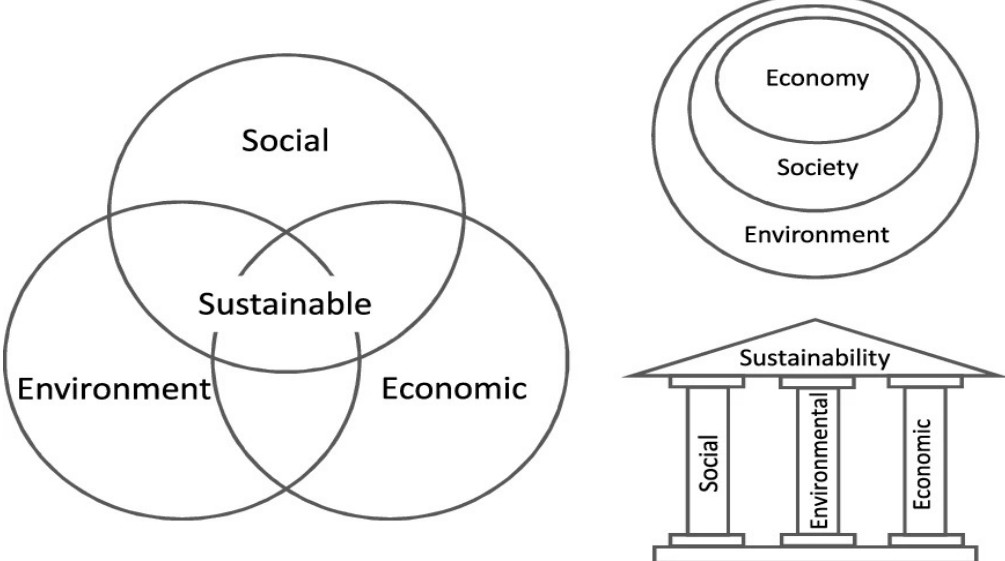

**Figure 3.** Three pillars and principles of sustainability.

One of the earliest critical public documents to use the three-domain model was *Our Common Future*, though this use is still largely implicit today. Ecology and economy were once distinct, conceptually and practically, but processes of globalization and growth have merged them. This merging of the two fields has created unique challenges in balancing human needs and environmental protection [45].

The demand for global interrelationships between the three domains has undoubtedly increased as expansion or development processes have ramped up. Ecologies, economies, and social relations have historically been intertwined in practice. Table 3 presents an overview of the criteria considered across different sustainability domains. The idea of sustainability as it relates to ecology, economics, politics, and culture—or even as it is used in one of those fields alone—is remarkably new [46].

**Table 3.** Domain-oriented principles and criteria.

| Environmental Domain | Social Domain | Economical Domain |
|---|---|---|
| Protect the health of the ecosystem | Social justice and equity | Adequate funds for social growth |
| Avoid excess pollution | Social infrastructure | Create employment and fair trade |
| Shift to renewable resources | Engaged governance | Rise the income of the people |
| Intergenerational decisions | Social capital | High standard of living |
| Target welfare, not GDP | Community and culture | Free and sharing market |
| Restoration and conservation | N/A | Cost saving and green finance |
| N/A | N/A | Financial stability and security |
| N/A | N/A | Green and circular economy |

The forty-year-old idea of sustainability is undergoing a paradigm shift in favor of The Dominant Domain Structure of Sustainability in the 21st century. We observe the same three-domain structure of sustainability in the various policy discourses linked to significant international organizations, with "the social" as a convenient term to group together those aspects of life that do not fall under the purview of the economy or the environment [47]. Since *Our Common Future*, three significant global initiatives have been launched. Each of these has impacted policy, administration, and sustainability governance, often directly influencing national and local engagement. The three major initiatives, the International Panel on Sustainable Development (IPCC), the Sustainable Development Goals (SDGs), and the United Nations Framework Convention on Climate Change (UNFCCC), are each instrumental in advancing global sustainability goals [48].

### 3.1.1. Political Domain

In order to envision and create a shared future, a deliberative political process is fundamentally required. In sustainability politics, resources are mobilized with an eye toward respect and harmony along multiple axes and over an extended period. Sustainability politics requires a unique approach to democratic deliberation that does not simply focus on immediate policy solutions but instead takes a broader and longer-term view of the impact of policies on society and the environment [44]. This approach enables us to identify and engage stakeholders, communities, and policy-makers in a shared vision of a sustainable future, allowing us to include all perspectives in the debate and reach equitable and effective decisions [44].

The topic of system regulation and governance identifies a fourth fundamental category of organization; the political sphere, which regulates the relations between (and within) the environmental sphere and the economic and social spheres. Establishing conventions, laws, and institutional frameworks for controlling society's social, economic, and, indirectly, environmental spheres creates the political sphere [49]. The political sphere serves as the "referee" who settles disputes involving the various, frequently incompatible claims made by participants in the social and economic spheres, both for themselves and concerning other spheres, such as the environment. This occurs through the political sphere as an intermediary, highlighting that, rather than direct environmental "governance", there is frequently an indirect connection between the political and environmental spheres [50]. The transition from politics to the environment may involve the "supply" of public policy meant to impact how environmental systems operate. Environmental–social then social–political integrations or environmental–economic then economic–political integrations are two ways to communicate societal demands "on behalf of the environment" [51]. Establishing conventions and procedures for regulating each sphere in relation to the others, to ensure the concurrent respect for quality/performance goals of all three spheres, constitutes the political or governance dimension of the organization [10]. This is the area of arbitrage between various principles and asserts of concern, obtained de facto or on purpose through coercion and institutional arrangements ranging from town and county councils to national government institutions to United Nations and other active international organizations. This type of regulation is a powerful force and creates a system of checks and balances that protects the interests of society and the environment. Such regulation works to ensure that all interests and needs, both those of individuals and larger groups, are taken into account to produce an optimal outcome for everyone involved [38].

### 3.1.2. Cultural Domain

In addition to art and literature, lifestyles, ways of interacting, value systems, traditions, and beliefs, culture is defined as the collection of unique spiritual, material, intellectual, and emotional characteristics of a society or social group. This leads to the perspectives and the character of a person as well as a society [52].

Along with sustainable development's environmental, social, and economic dimensions, culture has gained increasing attention in recent years. The protection of ecology and

the environment is motivated by the environmental dimension. The economic dimension encourages the effective use of financial resources and aims for long-term benefits, whereas the social dimension concentrates on the needs of humans, both present and future [53]. Since culture was previously included in the social dimension of sustainability, it was not considered a separate dimension. The situation gradually changed, though culture is now acknowledged as crucial to achieving sustainable development [27]. A separate cultural dimension of sustainability has been established to focus on the protection of traditional values and lifestyles, as well as the preservation of tangible and intangible heritage [52]. This separation of culture from the social dimension of sustainability is significant due to globalization, which has led to increased displacement and destruction of traditional cultures. Furthermore, this cultural dimension of sustainability includes respect for cultural diversity and promoting intercultural dialogue [54].

### 3.2. SDGs/MDG Linkages with Sustainability Domains

The idea of sustainability has garnered attention on a global scale and has been thoroughly discussed by academics, professionals, and decision-makers [55].

Several development goals and policies have been established to follow a sustainable vision and mission with sustainable development plans for stakeholders and the correct direction for our future survival [56]. The Millennium Development Goals (MDGs) were first established; they include eight main goals and address environmental, social, and economic challenges. MDGs, particularly, improved mortality, public health, hunger, and poverty [57]. However, all over the world remain a significant number of problems and challenges [58].

The Millennium Development Goals (MDGs) are significant and affect the international mobilization strategy for addressing several crucial global socioeconomic concerns. They convey mass political concern about gender inequity, ecological degradation, hunger, malnutrition, poverty, and illness [59]. The MDGs strive to advance awareness and understanding, political responsibility, enhanced measures, excellent interpersonal communication, and social opinion by condensing these objectives into a manageable group of eight goals and setting measurable targets within a limited period [60]. Developing nations have made significant strides toward achieving the MDGs. However, advancement rates vary greatly between targets, countries, and areas. The likely gap in MDG achievement is grave, sad, and highly unpleasant for low-income people. Yet, there is a widespread feeling among policy leaders and civil society that progress against poverty, hunger, and disease is significant; that the MDGs have played an essential part in securing that progress [61]. The shortage results from operational errors affecting numerous parties in wealthy and poor nations. Rich countries, for instance, have not complied with their promises of formal development support [62].

The Sustainable Development Goals (SDGs) are a group of 17 worldwide objectives to change the world. They are a component of the 2030 Agenda for Sustainable Development, created as a "Structured Road map to drive into a more equitable and sustainable future of planet and humanity" [63]. Each of the 17 goals is interconnected and addresses domains of sustainability in the present scenario while simultaneously reducing challenges such as poverty and the effects of climate change worldwide. The SDGs are intended to "defend the earth and enhance the lives and aspirations of everyone worldwide", according to the UN [64].

The 17 SDGs generally aim to satisfy the following summarized visions and objectives.

- Healthy life without hunger, malnutrition, and poverty [65].
- Ensure that everyone has universal access and the opportunity to utilize vital amenities such as sustainable energy, water, and sanitation [66].
- Encourage the creation of development possibilities through equitable employment as well as quality and accessible education.
- Promote innovation and robust infrastructure to build towns and cities that produce and consume things sustainably.

- Lessen global inequalities, particularly those related to gender equality and discrimination. Supporting and protecting weaker sections all over the world [67].
- Protect the marine and terrestrial ecosystems while battling climate change to preserve ecological integrity and the survival of the planet [68].
- Encourage cooperation amongst various social actors to foster a peaceful atmosphere and assure ethical production, trade, and consumption.

The objectives can only be met if they are incorporated into every aspect of government. Due to the complementarities, achieving one aim may aid in attaining others at the same time. Consider how tackling climate change challenges could enhance energy security, human health, ecosystems, and marine health. The main characteristic of the SDGs is that they are not standalone goals. Most goals are interconnected and interdependent and are well-defined in their perspectives and their plan of action of applications. Interconnectedness implies that achieving one goal leads to supporting another; therefore, they should be seen as connecting frames of a holistic and harmonic prominent structure. This is the key feature of the 2030 Agenda for Sustainable Development, which was adopted in 2015 by the United Nations [69]. There are 17 SDGs and recommendations for collaborative relationships among and within people, policy-makers, and other stakeholders worldwide. They address social, economic, and environmental concerns and promote sustainable perspectives from a broader viewpoint [58]. Hence, to pursue a sustainable world, the focus will be global collaboration and reducing inequalities and discrimination inside the states. Accordingly, all countries must work together to create a unified plan for implementing these goals, which can only be accomplished if each country accepts accountability for its conduct [23].

The initiative that led to the creation of the future Global Goals was more comprehensive, with policy-making governments incorporating corporates, communities, NGOs, other stakeholders, and individuals right from the start. We all must move toward the same direction of sustainability to attain global goals. It will take an extraordinary effort from all facets of society to realize these goals, and business must play a significant part in that endeavor [70].

Evidence shows that all earlier objectives are closely related to environmental challenges and sustainability issues [71]. More specifically, environmental justice and sustainable development will only function to the extent of their weakest SDG, much like the adage "a chain is only as strong as its weakest link" [72]. As an illustration, significant changes in the water, energy, and food sectors will be necessary for climate change mitigation, which is also essential to safeguarding the welfare of people. In other words, environmental justice and sustainable development goals must all be addressed effectively in order to achieve the full range of objectives for global sustainability [73].

## 4. Sustainability Challenge and Nexus

The interpretation of definitions and concepts of sustainability and sustainable development is still incomplete, and the perspectives are different for stakeholders in various domains and spheres. Even though sustainability addresses the relation and harmony between environmental, economic, and social spheres more extensively, executing the concept in the present scenario must be very specific and introduce a holistic approach. The ecological and biological aspects must be considered to accomplish the sustainability goals, which have not yet been met. To get rid the planet of anthropogenic problems, an integrated approach is urgently and consistently needed [74]. Figure 4 illustrates the various sustainability challenges.

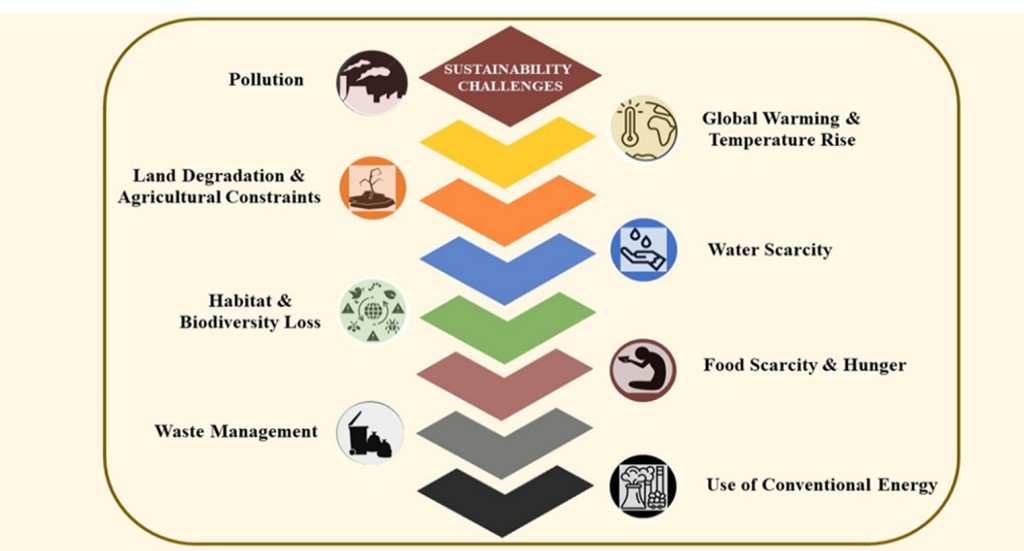

**Figure 4.** Sustainability challenges.

### 4.1. Pollution

The emergence of various pollutants over the past few decades as a result of human activity has had a negative impact on ecosystems. Due to rapid industrialization, rapid deforestation, and urbanization, all harming the ecosystems, conditions in developing countries are more dire [75]. Urban areas have developed into congestion hotspots, endangering mobility, air quality, water, and soil [76].

Due to the harm that plastics cause to ecosystems worldwide, it is a terrible ecological and environmental problem. Oil spills are another significant anthropogenic activity that harms the world's marine ecosystems [77].

Environmental pollution is a problem that affects both developed and developing nations, so it is a concern that is felt everywhere in the world. Researchers and decision-makers are eager to learn about the causes and effects of environmental pollution to develop potential remedies because it makes the Earth uninhabitable for living things and significantly contributes to global climate change. According to researchers, if environmental pollution persists, many regions will be covered by water while others will turn into deserts. In addition, extreme temperatures may exist everywhere [78]. The primary goal of the modern world is to combat environmental pollution by taking practical steps to safeguard those of us who live on the planet while being careful not to upset ecological balances. People must start reducing their waste and implementing environmentally friendly behaviors such as recycling and composting if they want the planet to remain habitable [44]. Additionally, people must avoid using too much energy or water from the environment [79]. Assisting citizens in lowering their carbon footprints, launching awareness campaigns, and implementing new laws that will lessen the effects of climate change are all things that governments should be implementing as well [77].

### 4.2. Global Warming and Temperature Rise

GHGs such as carbon dioxide ($CO_2$), methane ($CH_4$), nitrous oxide ($N_2O$), water vapor ($H_2O$), hydrofluorocarbons (HFCs), perfluorocarbons (PFCs), and sulfur hexafluoride (SF6) are among those contributing to the current climate change [80].

Extreme temperature changes over various parts of the Earth are predicted to happen sooner. With significant regional variations, the average global surface temperature has risen by 0.74 °C since the late 19th century and is projected to grow by 1.4–5.8 °C by 2100 [81]. A rise in sea level, changes in the distribution of plants and animals, increased environmental degradation, and natural disasters are all consequences of climate change. Other effects include hot weather, melting glaciers, polar warming, coral reef bleaching,

extreme precipitation events, prolonged droughts, and dry periods [82]. Thus, it can be concluded that, when it comes to the adverse effects of human activity, global warming and climate change are at the top of the list and must be addressed right away to restore the balance. Biological alternatives and solutions can also play a significant role in addressing climate change [83].

### 4.3. Land Degradation and Agricultural Constraints

Land degradation is a problem that affects all life forms and is not just limited to the deterioration of soil quality. Climate change is a significant factor that contributes to land degradation, as well as to a decline in the fertility and productivity of agricultural lands [84].

As a result of erratic precipitation patterns and rising global temperatures, severe weather events such as droughts and floods have also become more frequent, aggravating wind or soil erosion [85].

Another threat that is affecting more and more parts of the world is drought. Frequent cycles of drought and flooding have been brought on by climate change, and long stretches of water scarcity have led to desertification. Droughts have increased significantly over the past 40 years, particularly in the tropics and subtropics. The world has been experiencing water stress and environmental problems since the Anthropocene era, primarily attributed to human activity. One of the most noticeable effects of drought is on nutrient uptake because water is the medium by which nutrients are transported in plants [86]. Due to the unevenly distributed C, N, and microbial diversity caused by these adverse effects, the soil becomes infertile over time. In addition to biodiversity loss, wildfires, and soil erosion, drought harms habitats.

With an estimated surface area of 1 billion ha, soil salinization is another significant factor in land degradation affecting most countries. The drought and salinization of lands are related. The main contributors to land salinization have been agricultural practices and ineffective irrigation techniques. Low-quality irrigation water causes salt to build up in the soil, and poor drainage only worsens the situation [87].

Floods degrade land quality, disrupt agroecosystem productivity, and disturb vegetation. Since the 1950s, floods have become more frequently related to climate change's effects [88]. Waterlogging-induced hypoxia in plants results in poor root development, which reduces nutrient and water uptake and stomatal conductance, which results in wilting and decreased productivity [89].

### 4.4. Habitat and Biodiversity Loss

The disappearance of biological diversity has grown into a complex and ongoing issue. Due to this, biological heterogeneity has decreased, which has led to an unprecedented decline in terrestrial and marine species, including flora and fauna, affecting the ecosystems' overall stability. The main issue is the extinction of plant species because they are crucial to keeping the ecosystem balanced and directly impact how it functions by providing a habitat for various other organisms [90]. Although extinction is a natural occurrence, there is no denying that a wide range of human activities also contributes to the loss of biological diversity. Over the past 40,000 years, extinctions due to human activity have increased. Within the next 240 years, the Earth will likely experience its sixth mass extinction if current trends hold [87]. In fact, some researchers contend that anthropogenic activities alone are to blame for the beginning of the sixth species mass extinction. Estimates indicate that artificial habitats have replaced natural vegetation on 43% of the Earth's land surface [88]. This century, the rate of extinction is predicted to rise by a factor of two; with accelerated climate change, this rate may grow even faster.

The main threats to biodiversity are habitat loss and fragmentation, which are mutually dependent. Due to the population explosion, there is an increased demand for resources, which has resulted in the degradation of natural habitats and a severe threat to the habitats of plants and animals [91]. Examples of this demand include the expansion of

cattle ranching, mining, and building infrastructure. However, in addition to this, other anthropogenic activities are also significantly reducing the diversity of life on Earth.

### 4.5. Water Scarcity

Water resources are under pressure due to the ever-growing water demand brought on by population growth, economic development, and dietary changes. The World Economic Forum ranked the water supply crisis as the significantly higher risk facing our times [92].

It is crucial to comprehend water scarcity to create global, regional, national, and local policies. The Panta Rhei program set up a focused working group on "Water Scarcity Assessment: Methodology and Application" to create a cutting-edge methodology and assess water scarcity [93].

The northern hemisphere's middle-to-low latitudes generally have a high level of water scarcity, according to all the indicators. In almost all African nations, there is a severe problem with water scarcity, water poverty, and physical and economic water stress [94]. In order to retain objectivity and simplicity, all other water scarcity indicators created to date have been based solely on the physical quantity of water availability and use. Therefore, it is necessary to develop an integrated and consistent water scarcity assessment that simultaneously combines the physical, economic, and social aspects of water [95].

### 4.6. Food Scarcity and Hunger

The possibility of ending hunger by 2050 becomes doubtful with steady population growth. Hunger and malnutrition are primarily brought on by natural disasters, armed conflicts, population growth, and poverty [96]. The environment, the finite supply of food on the planet, and energy resources will not significantly impact these dynamics [97]. Therefore, while increased agricultural production and food preservation are essential to providing enough food for all, a more comprehensive approach is necessary to tackle the problem of global hunger [98].

### 4.7. Waste Management

Ecosystems and human health are seriously at risk due to the volume and complexity of the waste produced by the modern economy [99].

Waste management is also conducted to recover resources from the materials and lessen its environmental impact. Waste management can involve solid, liquid, or gaseous materials, and there are various techniques and procedures for each. Waste is managed using a variety of techniques, such as avoidance and reduction, energy recovery, recycling (physical and biological processing), and disposal (landfilling and incineration) [100]. The SDGs were created with this fundamental principle in mind. It is crucial to offer a comprehensive strategy based on sustainability as a concrete ideology that must be adopted as a way of life, a theory for formulating policies, and a sociopolitical concept of development to address the challenges the world is currently facing [101].

### 4.8. Industrialization and Sustainability Nexus

Industrialization refers to the economic and social change process that transforms a human group from an agrarian society into an industrial one. Regarding sustainability, industrialization has often been associated with negative impacts on the environment, natural resources, and human health and well-being. These negative impacts include pollution, deforestation, and the depletion of natural resources [102]. The nexus between industrialization and sustainability refers to the interconnected relationship between economic development, social well-being, and environmental protection. Sustainable development aims to achieve economic growth and improve living standards without compromising the ability of future generations to meet their own needs [103]. Industrialization can contribute to sustainable development by providing economic growth, jobs, and improved living standards. However, it also has the potential to harm the environment, deplete natural resources, and exacerbate social inequalities. Therefore, it is crucial to consider industri-

alization's environmental and social impacts and implement policies and practices that promote sustainable production and consumption [104].

### 4.9. Urbanization and Sustainability Nexus

Urbanization and sustainability are closely connected as urban areas are significant economic growth and development drivers, but they also have significant environmental and social impacts. Rapid urbanization can increase energy consumption, greenhouse gas emissions, environmental degradation, social inequality, and poverty. Urbanization can promote sustainable development by fostering compact, efficient, green cities [105]. Urbanization can impact the environment, including increased energy consumption and greenhouse gas emissions, water demand, and waste generation. However, sustainable urban development can help to mitigate these impacts through the promotion of energy-efficient buildings, the use of renewable energy sources, and the implementation of sustainable transportation systems. A nexus between urbanization and sustainability. The nexus between urbanization and sustainability refers to the interconnected relationship between the process of urbanization and the goal of sustainable development [79]. Urbanization can significantly impact the environment, economy, and society, and sustainable urban development is necessary to balance these impacts and promote liveable, resilient, and low-carbon cities. Urbanization can increase energy consumption, greenhouse gas emissions, and environmental degradation. It can also exacerbate social inequalities, poverty, and housing affordability issues. Sustainable urban development promotes social sustainability by addressing social equity, affordable housing, and access to services and job opportunities. It also supports the development of resilient cities that can adapt to the impacts of climate change and natural disasters. Sustainable urban planning is key to promoting sustainable urban development [106]. This includes promoting compact and efficient land use patterns, protecting natural areas and biodiversity, and integrating green spaces into the urban landscape. Additionally, sustainable transportation systems, such as public transportation, walking, and cycling, can reduce dependence on personal vehicles and improve air quality.

### 4.10. Globalization and Sustainability Nexus

Globalization and sustainability are related concepts that positively and negatively impact economic, social, and environmental development. On the one hand, globalization can lead to increased economic growth and improved living standards, greater access to goods and services, and enhanced communication and cultural exchange. However, it can also lead to adverse environmental impacts such as increased greenhouse gas emissions, biodiversity loss, and increased dependence on non-renewable resources [107]. Globalization can also exacerbate social inequalities, increasing poverty and marginalizing certain groups. The nexus between globalization and sustainability is complex, and it requires a holistic approach that considers the interrelated economic, social, and environmental aspects of global development. This can be achieved by implementing sustainable development policies, regulations, and initiatives that promote environmentally and socially responsible economic growth. One example of this nexus is sustainable trade, which supports economic growth while also addressing environmental and social concerns [108].

Globalization refers to the interconnectedness and interdependence of countries and economies by exchanging goods, services, information, and ideas. Globalization can lead to increased economic growth and improved living standards, but it can also contribute to environmental degradation and social inequality. On the other hand, sustainability aims to address these negative impacts by promoting environmentally and socially responsible economic development. Therefore, it is vital to find ways to balance the benefits of globalization with the need for sustainable development. This can include implementing policies and practices that promote sustainable production and consumption, protecting natural resources, and reducing inequality [107]. Globalization and sustainable development are related concepts that positively and negatively impact economic, social, and environmental aspects of development. On the one hand, globalization, which refers to the increased inter-

connectedness and interdependence of the world's economies, cultures, and populations, can lead to increased economic growth and improved living standards, greater access to goods and services, and enhanced communication and cultural exchange. Globalization can also lead to adverse environmental impacts such as increased greenhouse gas emissions, biodiversity loss, and dependence on non-renewable resources [109]. It can also exacerbate social inequalities, increasing poverty and marginalizing certain groups.

### 4.11. Climate Change and Sustainability Nexus

Climate change refers to the long-term changes in temperature, precipitation, wind patterns, and other measures of climate that occur over several decades or longer. It is primarily caused by burning fossil fuels, deforestation, and other human activities, which release greenhouse gases into the atmosphere, trapping heat and warming the planet. Addressing climate change is, therefore, a key component of sustainable development. This can include reducing greenhouse gas emissions by increasing the use of renewable energy, improving energy efficiency, and implementing carbon pricing [74]. Additionally, adaptation measures such as building sea walls or drought-resistant crops can help communities and ecosystems cope with the impacts of a changing climate. Therefore, it is essential to take action to reduce greenhouse gas emissions and to adapt to the changes that are already happening and those that are projected to occur in the future. Climate change can negatively impact the balance of global society by increasing poverty, reducing access to food and water, and exacerbating health problems [110]. On the other hand, addressing climate change through sustainable development practices such as renewable energy, energy efficiency, and sustainable land use can create economic opportunities and improve the well-being of communities.

The United Nations Framework Convention on Climate Change (UNFCCC) and the United Nations Sustainable Development Goals (SDGs) are closely related and mutually reinforcing. They provide a roadmap to achieve a better and more sustainable future for all. The Paris Agreement, adopted under the UNFCCC, aims to strengthen the ability of countries to address the impacts of climate change and to accelerate and intensify the actions and investments needed for a sustainable low carbon future. Sustainability plays a crucial role in addressing climate change. As previously mentioned, climate change is a significant threat to sustainable development, and addressing climate change is a critical component of sustainable development. Sustainability is vital in building resilience to climate change through disaster risk reduction and community preparedness [111]. Addressing climate change by promoting sustainable consumption and production patterns is desirable. This includes reducing waste, conserving natural resources, and promoting sustainable products and services [105]. To effectively face the challenges to sustainability due to climate change, a combination of mitigation and adaptation strategies, as well as international cooperation, access to finance and technology, and strengthened governance and institutions, are required.

### 4.12. Natural Disasters and Sustainability Nexus

Natural disasters can have a significant impact on sustainability. They can cause loss of life and injury, damage infrastructure and buildings, and disrupt economic activity. Additionally, natural disasters can exacerbate poverty, inequality, and vulnerability, particularly among marginalized communities. Climate change is projected to increase the frequency and intensity of many natural disasters, such as floods, droughts, and storms. This makes sustainability even more important as it can help to build resilience to these events [112].

Sustainability can help to reduce the risks and impacts of natural disasters in several ways [113]. Disaster risk reduction: By identifying and addressing the underlying risk factors that make communities and infrastructure vulnerable to natural disasters, sustainability can help to reduce the likelihood and severity of these events. Adaptation: Sustainability can help to build the resilience of communities and ecosystems to the impacts of natural disasters through strategies such as making sea walls, drought-resistant

crops, and community-based adaptation programs. Sustainable land use: Sustainable land use can help to reduce the risk of natural disasters such as floods and landslides by preventing deforestation and protecting wetlands and other natural habitats. Sustainable infrastructure: Sustainable infrastructure, such as green buildings and resilient transportation systems, can help to reduce the impacts of natural disasters on communities and economies. Community-based approach: A community-based approach to sustainability can ensure that the most vulnerable communities are involved in the planning and implementation of disaster risk reduction and adaptation measures. In conclusion, natural disasters can significantly impact sustainability and climate change is projected to increase the frequency and intensity of many natural disasters [114]. Sustainability can help to reduce the risks and impacts of natural disasters by building resilience, protecting vulnerable communities, and promoting sustainable practices.

*4.13. Population Rise and Sustainability Nexus*

Population growth can significantly impact sustainability, both positively and negatively. On the one hand, a larger population can lead to increased economic growth, technological innovation, and cultural diversity [106]. On the other hand, a rapidly growing population can strain natural resources and lead to environmental degradation, urbanization, and increased demand for energy, food, and water. One of the main concerns regarding population growth is its impact on the environment [115]. As the population increases, so does food, water, and energy demand. This can lead to the overconsumption of natural resources, deforestation, and pollution. Additionally, population growth can lead to urbanization and land-use changes, which can cause a loss in biodiversity and ecosystem services.

It is essential to consider sustainability in population growth control and development policies to address these challenges. This can include family planning programs, education and healthcare access, and policies promoting sustainable consumption and production. Promoting sustainable urbanization, land-use planning, and investing in renewable energy and water conservation technologies is also essential. Population growth and sustainable development are closely linked, as controlling population growth can significantly impact the ability to achieve sustainable development goals [116]. To address these challenges of population growth, it is crucial to incorporate population dynamics into sustainable development policies. This can include family planning programs, education and healthcare access, and policies promoting sustainable consumption and production. Promoting sustainable urbanization, land-use planning, and investing in renewable energy and water conservation technologies can also be included [103].

**5. Models and Principles of Socio-Economic Growth**
*Current Scenario*

In the 21st century, the ideology of sustainability and sustainable development has been globally accepted and gained momentum. While our understanding of sustainability has significantly increased, development has become harder to define in many ways. However, studying sustainable development is insufficient because it is time to act.

In this post-Brundtland era, sustainability, through the Sustainable Development Goals, addresses challenges such as increasing conventional energy consumption, loss of biomass and land degradation, skepticism of science, financial disparities in life and opportunities, and a fragmented set of universal policies, institutional frameworks, and governance. Furthermore, due to several interconnected phenomena, the difficulties of sustainability and development are more complex today than they were in Brundtland's time. Adopting pluralistic and transdisciplinary approaches to sustainability analysis is a crucial strategy to face the challenges of the present scenario around the globe.

The contentious nature of sustainability as a dominant policy discourse has encouraged the formation of many public forums for discussion and engagement [117]. Though idealistic, the concepts and methods point to the fact that the deliberative democracy, which

includes open discourse, open decision-making, holding decision-makers accountable, as well as reasoned and respectful debate, is essential to achieve green development in public spheres wherein the various sustainable development ideas can be discussed and improved upon and manage it socially, politically, and financially [118].

The task of defining sustainability and sustainable development has been a complex and ongoing endeavor for researchers. The lack of a universally accepted definition stems from the diverse interpretations attributed to the phrase, particularly in relation to its association with "economic growth". This has sparked a debate among scholars, as some argue that traditional notions of development, synonymous with continuous economic expansion, are incompatible with sustainability, given the finite resources of our planet [119]. Bringing together the diverse perspectives of sustainability under a unified framework has proven to be a challenging task. Over the years, the definitions of sustainability have evolved while retaining their core essence [105]. However, there is still a need for a comprehensive explanation that can encompass all the different domains of sustainability. Additionally, it is important to acknowledge that the concept of sustainability is influenced by the economic growth and political ideologies prevalent in different nations. Table 4 presents a comparative analysis of different socio-economic growth models.

**Table 4.** Comparison of capitalism, socialism, and communism.

| | **Pros** | **Cons** |
| --- | --- | --- |
| Capitalism | Most efficient and effective way to allocate resources | This leads to economic inequality, environmental degradation |
| | Create enormous wealth | Focus on profit over the well-being of people and communities |
| | Encourages innovation and hard work. | Different levels of government regulation and intervention |
| Socialism | More just and fair economic system | Lead to inefficiencies and a lack of financial incentives |
| | Reduce income inequality and provide a safety net for all | Government has a low level of control over the economy and the lives of its citizens |
| | Focus on the well-being of people and communities | It can limit individual freedom and personal responsibility |
| Communism | Meeting the basic needs of all members of society and maximizing the collective well-being | Economic inefficiency and widespread human rights abuses |
| | Seeks to eliminate the exploitation of one person by another | |
| | Create a society based on equality and cooperation | |

Capitalism: Capitalism is an economic system in which the means of production, prices of commodities (goods and services), and distribution are privately owned and operated. The prices are determined by supply and demand in a competitive market [120]. In a capitalist economy, the goal is to make a profit, and businesses are free to operate and compete with one another [121].

Socialism: Socialism is an economic and political system in which the means of production and distribution are owned and controlled by the state community rather than private individuals [122]. In a socialist system, the wealth produced by the economy is shared more equally among the members of society, and there is often a strong emphasis on providing for the basic needs of all people, including healthcare, education, and social security [123].

Communism: Communism is a political and economic ideology that seeks to create a classless, stateless society in which the means of production and distribution are owned and controlled by the community as a whole [124]. Most generally, communism refers to community ownership of property, with the end goal being complete social equality via economic equality. Under communism, the goal is to create a society where everyone works according to their abilities and receives according to their needs [125]. Fundamentally, communism argues that all labor belongs to the individual laborer; no man can own another man's body, and therefore each man holds his work.

The universally accepted capitalistic GDP growth-driven economic model, which has prevailed for the last century, has proven to be a complete failure and is leading us toward a disastrous path. Reliant on fossil fuel burning, this model exacerbates climate change, pollution, biodiversity loss, and freshwater depletion while reinforcing unequal wealth distribution. Its promises of economic success and improved quality of life for citizens have come at the expense of the environment and social equity. In summary, these socio-economic models have their drawbacks and challenges, and there is no one-size-fits-all solution for achieving sustainable economic growth and development. Urgent and comprehensive revaluation is needed to shift toward alternative mechanisms prioritizing sustainable development, incorporating concepts such as the circular economy, nature-based solutions, social innovation, and responsible consumption and production patterns, to ensure a resilient and equitable future for all. A more holistic and inclusive approach must be identified and developed to create a more sustainable and equitable future.

## 6. Global Sustainability and Sustainalism: An Integrated Framework

The world's economic development model has become saturated with an excessive focus on increasing consumption. As a result, humanity is confronted with many significant threats, including climate change, health crises such as COVID-19, and economic instability. These crises have served as powerful reminders of the importance of cooperative actions and global solidarity. Therefore, the path to global sustainability lies in educating the masses and nurturing a knowledge-based economy and socially responsible society [126]. Sustainalism builds on the foundations laid by capitalism and socialism but takes the broader view that the challenges of today and tomorrow demand of us. The more considerable paradigm shift from capitalism, communism, and socialism is sustainalism. We need a paradigm shift from capitalism or moderated socialism [127]. The new model of social economy for the current generation is referred to as "sustainalism" [128].

### 6.1. Global Sustainability 6S Principles: A Tool to Achieve a Sustainable Economy

It is crucial to recognize that all life forms on Earth, including humans, animals, and plants, are intricately interconnected. In the era of globalization and rapid digitalization, countries worldwide, regardless of their size, wealth, or level of development, rely on each other in various aspects. This interconnectedness is fostered through economic, cultural, and social relations. The scientific community comprehends the inherent value of this interconnectedness and its implications for our collective well-being. Furthermore, the distribution of global resources is highly unequal, posing challenges to achieving sustainable development. A genuine appreciation for Mother Nature, the Earth, and its delicate ecosystems is at the core of sustainable actions. This sentiment serves as the foundation for sustainable practices. It drives individuals, organizations, and governments to adopt a holistic approach to development that integrates economic, social, and environmental considerations in every corner of the world.

A plan for a sustainable economy is presented in a simple equation format.

Sustainable Economy = 6S Principles of Global Sustainability
= Happiness + Well-being + Equality
= Regenerative Practices +Climate and Biodiversity Protection + Ecological Restoration

The "Global Sustainability Framework" is a novel comprehensive toolkit comprising 6S principles (Figure 5). This framework equips individuals, organizations, and governments with the tools to pursue global sustainability effectively. However, the responsibility for creating a sustainable future lies on the shoulders of global citizens, who must embrace this responsibility and work collectively to ensure a thriving and sustainable world for future generations.

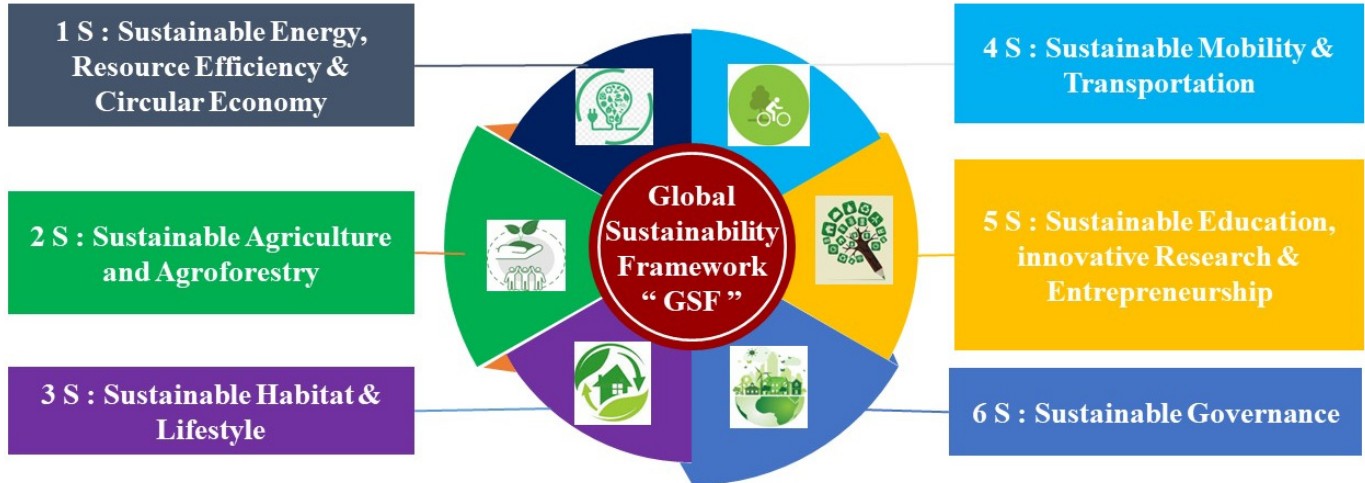

**Figure 5.** Tool for sustainalism (6S principles).

**1S—Sustainable Energy, Resource Efficiency, and Circular Economy:**

1S principle is a compelling approach highlighting the urgency of transitioning to sustainable energy sources, promoting resource efficiency, and adopting a circular economy. Our reliance on finite energy sources, such as fossil fuels, poses significant risks for future generations [129]. By embracing sustainable energy, we can mitigate these risks and ensure a more secure and stable energy supply for the long term.

The need for a sustainable energy transition is evident. Investing in renewable energy infrastructure and promoting energy efficiency measures are crucial in reducing greenhouse gas emissions and combating climate change [130]. Technologies such as solar panels, wind turbines, and hydroelectric power plants offer promising avenues for generating clean and renewable energy [131].

In addition to transitioning to sustainable energy sources, resource efficiency plays a vital role in promoting sustainability. We can effectively utilize water, materials, and energy resources across various sectors by optimizing resource consumption, minimizing waste generation, and encouraging recycling and reuse [132]. Energy efficiency improvements, in particular, offer significant opportunities to reduce energy consumption and enhance sustainability [133].

The concept of a circular economy further strengthens the 1S principle. By designing out waste and pollution, extending the lifespan of products and materials, and regenerating natural systems, the circular economy offers a transformative approach to resource management (Figure 6). It shifts the focus from a linear take–make–dispose model to one that values waste as a resource and prioritizes reuse, remanufacturing, and recycling [134]. Embracing circular business models and strategies promotes sustainability, reduces resource depletion, and minimizes environmental impact [135].

Adopting a circular economic model is crucial for overcoming the challenges posed by climate change and resource depletion [136]. By fundamentally rethinking our approach to resource utilization, we can break free from the unsustainable practices of the past and build a resilient and sustainable economy. This shift benefits the environment, presents economic opportunities, and fosters innovation in sustainable technologies and practices [137].

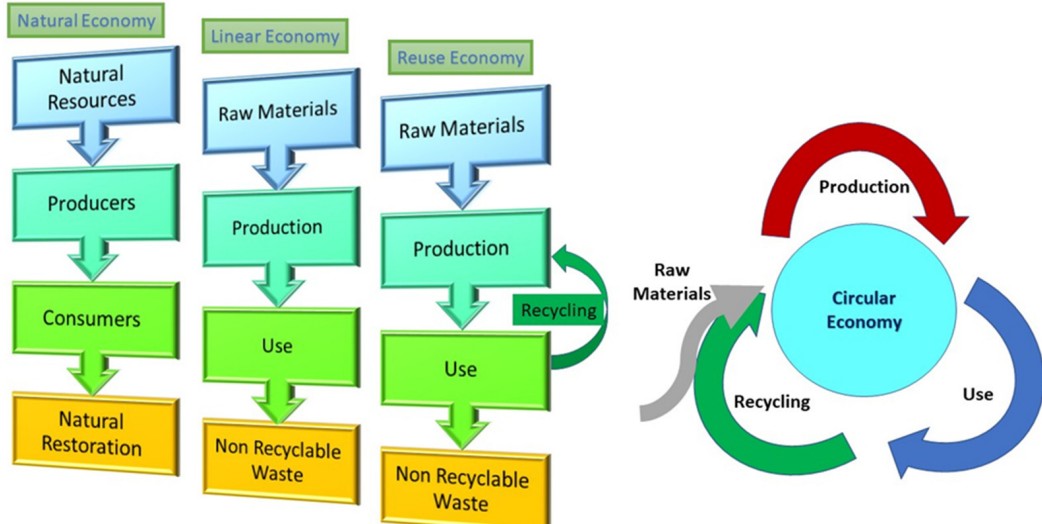

**Figure 6.** Concept of circular economy.

**2S—Sustainable agriculture, agroforestry, and bioeconomy:**

2S—Sustainable agriculture and agroforestry present a compelling solution to address food security challenges while promoting sustainability [138]. These practices go beyond conventional farming methods and integrate environmental, social, and economic considerations into agricultural systems [139].

Sustainable agriculture encompasses a range of practices prioritizing the conservation of natural resources, protection of the environment, and the well-being of farmers and communities. By adopting sustainable farming techniques, such as organic farming, precision agriculture, and crop rotation, farmers can minimize the use of harmful chemicals, preserve soil fertility, and reduce water consumption [140]. This approach ensures the long-term viability of agricultural land and safeguards the health of consumers and ecosystems.

Conversely, agroforestry combines agricultural activities with cultivating trees and woody plants in a mutually beneficial manner. This integrated approach offers numerous benefits for sustainable land management [141]. Trees and wood plants serve as natural allies, contributing to soil conservation by preventing erosion, enhancing soil health through increased organic matter, and improving water retention capabilities. They also play a vital role in conserving water resources by reducing evaporation and promoting water infiltration into the soil. Furthermore, the carbon sequestration potential of trees and woody plants helps mitigate climate change by absorbing atmospheric carbon dioxide [142].

Agroforestry systems also provide valuable habitats for wildlife, including birds, insects, and mammals [143]. Creating diverse and interconnected ecosystems contributes to biodiversity conservation and supports the preservation of valuable ecological services. Moreover, agroforestry allows farmers to diversify their income streams and improve economic stability by introducing a broader range of products and value-added opportunities [144].

Bioeconomy encompasses various production sectors, such as industrial and economic domains, which utilize biological resources and methods to produce bio-based goods and services [145]. The bioeconomy concept revolves around achieving sustainability by leveraging natural resources and processes in economic activities [146]. In doing so, it fosters the development of bio-based industries and generates employment opportunities [147]. A regenerative economy and bio-economy provide the opportunity to fulfill the needs of all individuals emphasizing the restoration, regenerative practices—embracing the potential of the bio-economy—and renewal of natural resources, thereby safeguarding the health and vitality of our planet.

We can address food security challenges by embracing sustainable agriculture and agroforestry while promoting environmental stewardship and social well-being [148]. These practices provide a holistic and resilient approach to agricultural production, ensuring the long-term availability of nutritious food, safeguarding natural resources, and supporting local communities.

**3S—Sustainable building, health, and lifestyle:**

Sustainable living is making environmentally, socially, and economically responsible choices to reduce one's impact on the planet and contribute to a more sustainable future [149]. This can involve making lifestyle changes, such as

Sustainable built environment: Incorporating sustainable transportation principles into urban planning and infrastructure development [150]. This includes designing cities and communities prioritizing walkability, cycling infrastructure, and efficient public transportation systems, reducing the need for long-distance commuting and promoting compact and sustainable development [150].

Green infrastructure/sustainable practices: Sustainable living often involves reducing one's consumption of resources, such as energy, water, and materials. This can be achieved through energy conservation, water conservation, and waste reduction [151]. It involves supporting businesses and organizations that adopt sustainable practices, such as using renewable energy, reducing waste, and protecting the environment [152]. Additionally, effective land use planning is vital in curbing and, ideally, preventing urban sprawl, contributing to the depletion of natural and agricultural lands. A sustainable lifestyle bestows positive effects on one's physical and mental health. The integration of sustainable building, health, and lifestyle creates a comprehensive approach to living in harmony with the environment while prioritizing personal well-being.

Protecting natural habitats: Sustainable living often involves protecting and preserving natural habitats, such as forests, wetlands, and oceans [153]. Implementing green infrastructure networks is crucial in filtering and purifying water and air while promoting energy and water efficiency through retrofitting measures in new and existing developments. By integrating human and environmental consciousness into our lives, we can make more sustainable choices that benefit both people and the planet [154,155].

**4S—Sustainable mobility, transportation, and eco-tourism:**

4S—Sustainable Mobility, Transportation, and Eco-Tourism offer a transformative approach to address sustainability challenges in transport and tourism. This comprehensive principle encompasses various elements that can revolutionize how we move and explore the world while minimizing environmental impact and fostering inclusive and responsible practices.

Sustainable Transportation Systems are crucial in reducing carbon emissions, alleviating traffic congestion, and improving air quality [156]. By promoting and prioritizing sustainable modes of transportation such as walking, cycling, public transit, and electric vehicles, we can create a more environmentally friendly and efficient transportation network [157].

Efficient and Integrated Transport Networks are essential for optimizing travel routes, enhancing connectivity, and reducing the overall environmental footprint of transportation activities [158]. By designing and implementing integrated transport systems that prioritize efficiency and sustainability, we can significantly improve the overall transportation experience [159].

Active and Shared Mobility encourages individuals to embrace operational modes of transportation such as walking and cycling while promoting shared mobility options such as carpooling and ride-sharing [160]. These initiatives help to reduce the reliance on private vehicles, decrease traffic congestion, and encourage sustainable travel choices.

Accessible and Inclusive Transport is vital to ensure that transportation systems cater to the diverse needs of individuals, including those with disabilities, the elderly, and those with limited mobility [161]. By designing infrastructure, vehicles, and services that are accessible and inclusive, we can create transportation systems that leave

no one behind [162]. Moreover, fostering sustainable supply chains and collaboration across stakeholders is crucial. This involves promoting responsible sourcing, reducing carbon emissions in transportation, and minimizing environmental impacts throughout the supply chain.

Eco-Tourism and Sustainable Travel promote responsible tourism practices that minimize negative environmental impacts, support local communities, and preserve natural and cultural heritage [163]. Emphasizing eco-friendly behaviors, supporting local economies, and raising awareness about sustainable travel choices can lead to a more sustainable and enriching travel experience [164].

By integrating these elements within the 4S principle, we emphasize the importance of sustainable mobility, transportation systems, and eco-tourism in mitigating climate change, enhancing accessibility, and preserving our natural and cultural resources. It calls for prioritizing sustainable travel choices, embracing efficient transportation modes, and incorporating sustainability into urban planning and tourism practices for a more sustainable and inclusive future.

**5S—Sustainable Education, Innovative Research, and Entrepreneurship:**

5S—Sustainable Education, Innovative Research, and Entrepreneurship form a dynamic and transformative approach to addressing sustainability challenges. This principle highlights the critical need for an education system that actively prepares individuals to contribute to sustainable development, fosters innovative research, and nurtures entrepreneurial endeavors for a sustainable future [165].

Sustainable Education is the cornerstone of this principle, advocating for educational systems that integrate sustainability principles at all levels, from primary to higher education [166]. By infusing environmental awareness, social responsibility, and sustainable practices into the curriculum, we can equip students with the knowledge and skills necessary to navigate and thrive in a sustainable world. Hands-on experiences, experiential learning, and lifelong learning opportunities further empower individuals to adapt to evolving sustainability needs and foster a mindset of continuous growth and development [167]. Education for sustainable development emphasizes critical thinking, problem-solving, and global citizenship, preparing future generations to address sustainability challenges.

Education and Outreach initiatives are crucial in raising awareness about the importance of sustainable energy and its various applications. By implementing public information campaigns and sustainability education programs, we can engage and inspire individuals to embrace sustainable practices and promote the adoption of sustainable energy solutions [168]. These efforts contribute to a broader cultural shift toward sustainability and encourage active participation in creating a more sustainable future.

Knowledge Transfer and Collaboration are fundamental aspects of sustainable education, research, and entrepreneurship. Facilitating knowledge and technology exchange between academia, industry, and communities creates synergies that drive sustainable development forward [169]. Collaborative partnerships leverage expertise and resources, fostering innovation and enabling the practical application of research outcomes. By establishing strong ties with local organizations and businesses, educational institutions can provide students with real-world exposure to sustainability challenges and opportunities, empowering them to impact their community [170] positively.

Entrepreneurship and Social Innovation are integral to addressing sustainability challenges effectively [171]. We foster interdisciplinary collaboration and propel sustainable development by supporting research endeavors that tackle these challenges and contribute to innovative solutions. Cultivating an entrepreneurial culture encourages individuals to generate ideas and develop solutions aligned with sustainability goals [172]. Providing aspiring entrepreneurs in the sustainable sector with the necessary support, resources, and mentorship enables them to translate their ideas into impactful ventures.

In the digital age, incorporating technologies such as digitalization, AI, IoT, and intelligent and automated systems further amplifies the potential for sustainable education, research, and entrepreneurship. These tools enhance efficiency, optimize resource utilization, and enable more effective decision-making, contributing to sustainable practices and outcomes. This perspective empowers individuals to become agents of change, equipping them with the knowledge, skills, and entrepreneurial spirit needed to address sustainability challenges and create a more sustainable and prosperous future for all.

**6S—Sustainable business, governance, and finance:**

6S—Sustainable governance is a powerful approach that addresses sustainability challenges by promoting a culture of sustainability, ethics, and responsible decision-making within organizations [173]. This principle recognizes the significance of integrating social justice considerations into sustainable organizational culture and ethical governance practices [174]. We can leverage various approaches to achieve sustainable governance by adopting a holistic perspective.

A sustainable organizational structure is crucial for long-term stability and effectiveness while ensuring environmental, social, and economic sustainability. Organizations must proactively design systems that align with sustainable principles, enabling them to adapt to changing circumstances and prioritize sustainability in their operations. This involves considering the environmental impact of business practices, fostering social responsibility, and optimizing economic outcomes sustainably [175].

Ethical Leadership serves as a cornerstone of sustainable governance. Leaders at all levels of an organization must embody moral values, champion social justice, and address systemic inequalities. By embracing diversity and inclusion, ethical leaders create an environment that values the contributions of all individuals and promotes a sense of fairness and justice [176]. Ethical leadership fosters a culture where sustainable practices are embedded into decision-making processes and guides the organization toward long-term sustainability goals.

Social Justice values are integral to sustainable governance. Organizations must embed equity, fairness, and social justice principles into their organizational culture and decision-making processes [177]. This entails promoting inclusivity, embracing diversity, and providing equal opportunities for all organization members. By aligning all levels of the organization around shared values such as sustainability and social justice, we create a foundation for sustainable practices and facilitate collective efforts toward long-term sustainability.

Sustainable Policy and Stakeholder Engagement play a crucial role in sustainable governance. Policy interventions, such as tax credits, incentives for renewable energy, energy efficiency regulations, and funding for research and development, encourage adopting sustainable practices [178]. Moreover, engaging stakeholders, including employees, communities, customers, and other relevant actors, allows for a collaborative approach to decision-making [179]. By involving diverse perspectives, organizations can make more informed and sustainable decisions that consider the needs and interests of all stakeholders [180].

Collaborative Decision-Making is essential for sustainable governance. Encouraging collaboration and participation from all levels of the organization enables a diversity of perspectives and ideas to be considered. We can foster sustainable collaboration and drive positive change by establishing common agendas, engaging in participatory decision-making, and monitoring progress. This inclusive approach facilitates the identification of innovative solutions, ensures transparency in decision-making processes, and fosters a sense of ownership and commitment among stakeholders. By embracing collaborative decision-making, organizations can effectively address sustainability challenges and promote adopting sustainable practices. Figure 7 provides a concise overview of the essential components of the 6S principles, which serve as a roadmap for attaining global sustainability.

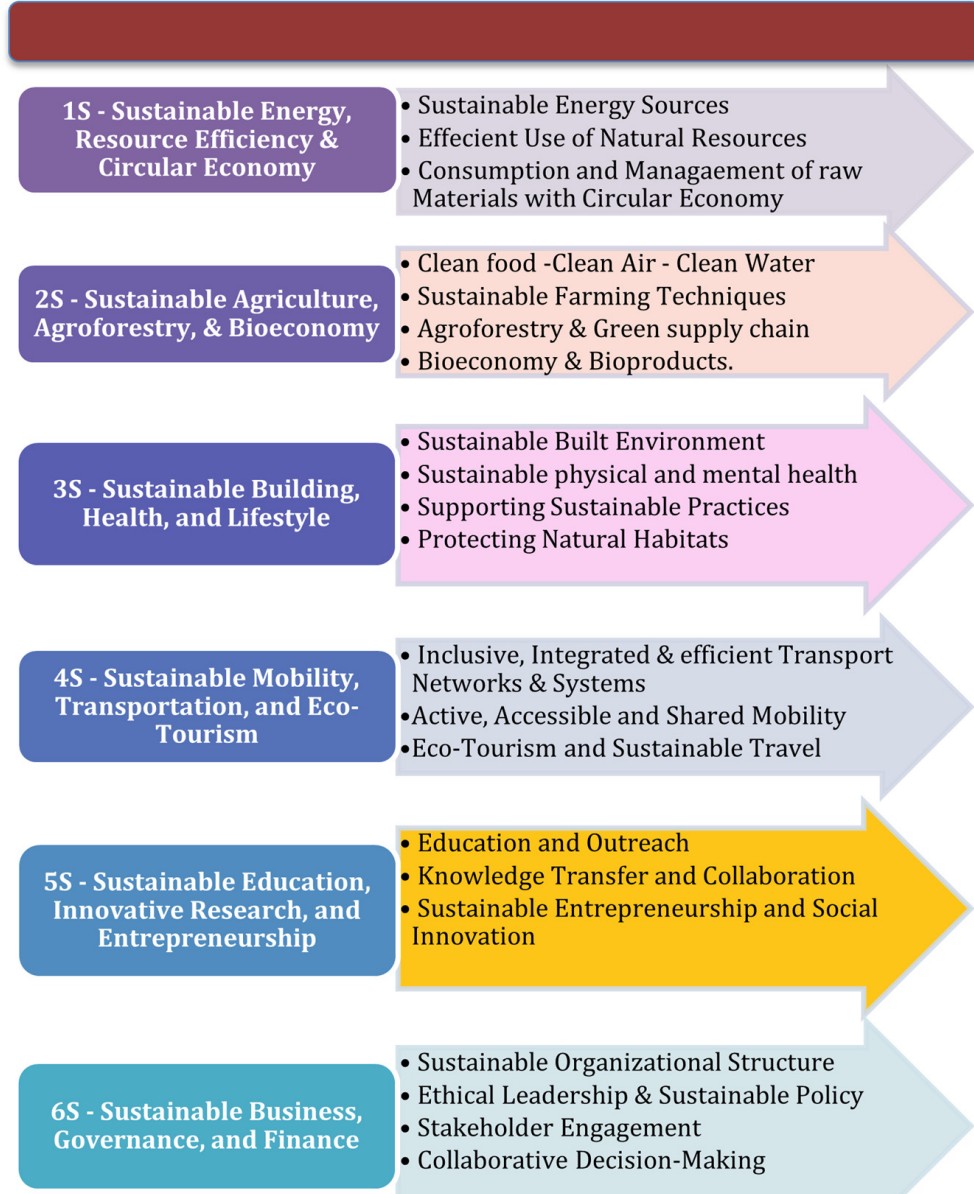

**Figure 7.** Global sustainability 6S principles: A tool to achieve sustainability.

By incorporating sustainable energy practices, resource efficiency, and circular economy principles (1S), we can reduce reliance on finite resources, mitigate climate change, and minimize waste. Sustainable agriculture and agroforestry (2S) contribute to food security, soil conservation, and habitat preservation. Sustainable mobility and transportation (3S) promote low-carbon options, reduce congestion, and enhance accessibility. Sustainable habitat and lifestyle (4S) focuses on sustainable urban development, healthy life style choices, physical and mental well-being, waste management, and responsible consumption. Sustainable education, innovative research, and entrepreneurship (5S) foster knowledge transfer, skills development, and solutions for sustainability challenges. Finally, sustainable governance (6S) promotes ethical leadership, social justice, and stakeholder collaboration.

By embracing the 6S—Sustainable governance principle, organizations can create a framework that fosters sustainability, ethics, and responsible decision-making. This approach integrates social justice considerations into organizational culture, promotes

ethical leadership, engages stakeholders, embraces collaborative decision-making, and leads to a more sustainable and equitable future.

### 6.2. Concept of Sustainalism

The term sustainalism has not gained widespread usage. It is not well-defined or widely recognized in the literature. "Sustainalism" is a socio-political, economic, and environmental theory of global social organizations as a whole that advocates for the means of production, distribution, exchange, and symbiotic lifestyle to be owned or regulated by the worldwide community holistically [127]. Sustainalism is a social equity and inclusiveness theory built on the foundations laid by capitalism, communism, and socialism (Figure 8). Sustainalism is the new, inclusive, and more equitable socio-economic–environmental theory and practice model of the 21st century to meet the needs of the 10 billion people who will share a single planet in just a few decades from now. Sustainalism is an Earth-friendly conscious civilization that inculcates a socio-economic–environmental model to decarbonize the physical economy and embrace green products and services. Sustainalism is a term that has been used to describe a concept or approach that combines elements of sustainability with elements of traditionalism [126]. Sustainalism is a resource-efficient lifestyle wherein the materials are economically produced from 100% natural resources such as wood, plant fibers, etc.

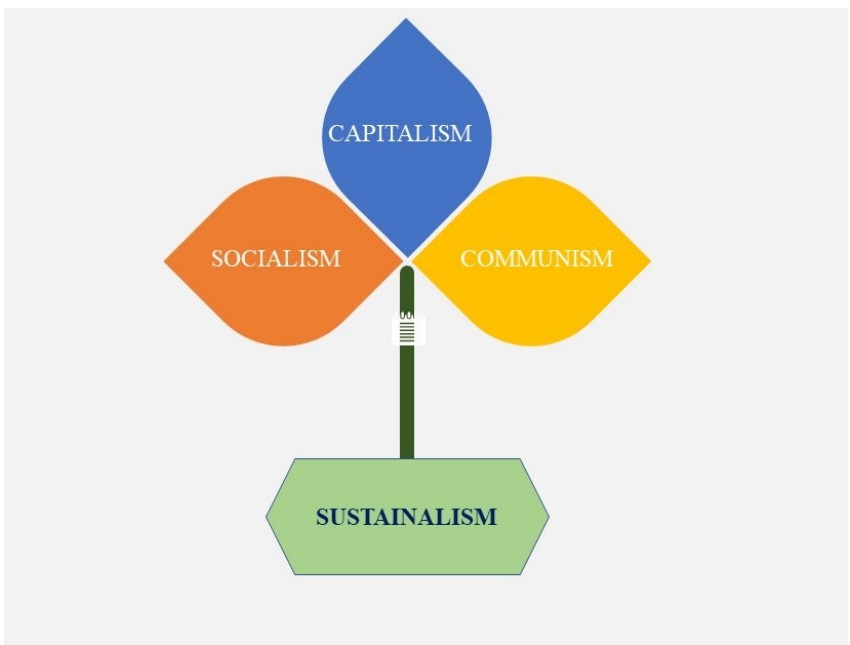

**Figure 8.** Concept of sustainalism.

Sustainalism is a new model of survival and a practice leading to a sustainable era. Sustainalism paves the path to a "Sustainable Revolution" which satisfies the needs of our future. Sustainalism is an art of social engineering toward a greener and more sustainable lifestyle while promoting harmonious relationships within the environmental, economic, social, political, and cultural domains (Figure 9). This includes adopting ecological principles, eco-lifestyle, decarbonizing the physical economy, as well as embracing green products and services [181]. Sustainalism is a collaborative practice of all the stakeholders, including governments, organizations, private sector, public sector, service sectors, corporates, entrepreneurs, investors, and individuals who are collectively responsible for shifting the global socio-economy from the current equilibrium, which is a status quo to a better, that is, cleaner, renewable, and sustainable planet [182].

## SUSTAINALISM AND DOMAINS

**Figure 9.** Domains of sustainalism.

### 6.3. Objective of Sustainalism

The objectives of sustainalism are as follows:

- To create economic growth and prosperity while protecting the environment and promoting social equality.
- To find ways to live and consume environmentally sustainably and maintain and preserve traditional cultural practices and values.
- To focus on local communities, self-sufficiency, and intergenerational equity.
- To complement education, leadership, and collective consciousness to sustain a quality life for society.
- To emphasize using nature-based solutions, such as green technology and carbon pricing, to address economic, environmental, and social problems.
- To advocate for creating new businesses, policies, and regulations that promote environmental, social, and economic sustainability in the short and long term.

### 6.4. Role of Individuals in Sustainalism: Sustainalist

"Sustainalist" is a term used to describe a person who prioritizes sustainability in all aspects of life, including environmental, economic, and social domains. It emphasizes the need to balance economic growth with environmental protection and social justice so that current needs can be met without compromising the ability of future generations to meet their own needs. The role of sustainalists in the future will be crucial in promoting and advocating for sustainable practices and policies that prioritize the well-being of current and future generations [183]. Sustainalists will work toward creating a more equitable and environmentally responsible world by promoting environmentally friendly technologies and practices, reducing carbon emissions, preserving natural resources, and practicing social justice and equality. They will also play a key role in raising public awareness about the importance of sustainalism, and influencing businesses, governments, and the public to adopt a new socio-economic–environmental model of sustainalism. By working toward a more sustainable future, sustainalists will help ensure that the planet remains habitable and that future generations can thrive. Becoming a sustainalist involves incorporating sustainable principles and practices into your daily life and advocating for policies and initiatives that prioritize sustainability [183].

Here are a few steps to becoming a sustainalist:

1.　Educate yourself: Learn about the principles of sustainalism, including environmental, economic, and social domains, and the impact of human activities on the planet.
2.　Reduce your carbon footprint: Start by reducing waste, conserving energy, and using environmentally friendly products.
3.　Support sustainable businesses: Look for products and services that prioritize sustainability and support companies that have environmentally friendly practices.
4.　Advocate for sustainable policies: Write to your local representatives, participate in environmental campaigns, and raise awareness about the importance of sustainalism.
5.　Live sustainably: Incorporate sustainable practices into your daily life, such as cycling or taking public transportation instead of driving, eating a plant-based diet, and conserving water.
6.　Lead by example: Encourage others to adopt sustainable practices by sharing your experiences and knowledge with family, friends, and colleagues.

By following these directions, we can take a step closer to realizing the principles of sustainalism.

### 6.5. Role of Society in Sustainalism

Sustainability, closely related to sustainalism, is a relatively new concept that has gained widespread recognition and support in recent years, and this trend will likely continue. However, the ideas and values associated with sustainalism will likely continue to be meaningful and influential. As the global population continues to grow and the impacts of human activity on the environment become more apparent, there will likely be increasing emphasis on finding ways to live and consume more sustainably. We have an exceptional opportunity for individuals, sectors, companies, and organizations to go down in history as the generations that changed the course of the world for the better [183]. This may involve a focus on local communities, self-sufficiency, intergenerational equity, and other values and practices associated with sustainalism. A sustainable lifestyle consists of promoting social equity, diversity, inclusion, social justice, fair labor practices, advocating for human rights and reducing inequality. Adopting a sustainable lifestyle requires a shift in our attitudes, values, and beliefs. It requires us to recognize our interconnectedness with each other and the natural world. It also requires us to take collective responsibility for our actions and their impact on the environment and society.

### 6.6. Role of Nations: Sustainable Revolution

Sustainable revolution refers to a significant transformation at the global level to achieve a more sustainable future [184]. It involves a shift in values, attitudes, and behaviors toward sustainalism and adopting sustainable practices and policies at all levels of society, from individuals to governments and various states [185]. The sustainable revolution aims to create a more equitable and environmentally responsible world wherein economic growth is balanced with environmental protection and social justice. "Sustainable Era" refers to a time in which sustainalism is the dominant paradigm, and sustainable practices and policies are widely adopted and implemented. A strong focus on environmental protection, resource conservation, and social equity and the widespread adoption of sustainable technologies and practices characterizes the sustainable era. In this era, economic growth is decoupled from environmental degradation, and the world operates in a way that prioritizes the well-being and survival of both current and future generations. The sustainable era aims to create a more livable and sustainable world for all.

## 7. Conclusions

In conclusion, sustainability has emerged as a crucial response to environmental degradation, social inequality, and economic instability. However, traditional approaches to sustainable development have proven inadequate in tackling these complex challenges, necessitating a more comprehensive and holistic approach.

To address these limitations and foster a paradigm shift toward a more sustainable and inclusive world, this paper proposes an integrated socio-economic and environmental model, the 6S principles of global sustainability. We have presented a novel perspective on achieving sustainable development goals through a social movement centered around sustainable education, sustainable living, peace, social justice, social equity, sustainable housing, sustainable networks (including mobility and health infrastructure), and sustainable energy. The 6S theoretical framework offers a clear roadmap toward achieving global sustainability and effectively tackles the challenges related to sustainability through an inclusive approach.

To enhance individual responsibility toward sustainable development, the concept of sustainalism is introduced. Building upon the principles of sustainalism, the Global Sustainability Framework recognizes the interconnectedness of different dimensions of sustainability and the diverse Sustainable Development Goals (SDGs). It highlights the importance of collective action, dedication, and collaboration among individuals, organizations, and governments for a fair and inclusive quality of life. Sustainalists adopt this new way of thinking and practice, recognizing the interdependence of all living beings and advocating for social and environmental justice.

Implementing the Global Sustainability Framework and embracing sustainalism necessitates a sustainable revolution—an unprecedented collective movement to reshape our societies, economies, and governance systems toward sustainability. The sustainable revolution offers a transformative pathway toward achieving an equitable world, marking a significant step toward a sustainable era.

By embracing the principles of sustainalism and adopting a sustainalist lifestyle, we can pave the way toward a more sustainable economy that balances humanity's and the environment's needs, benefiting everyone involved.

**Author Contributions:** Conceptualization, K.S.; formal analysis, K.B.M.; investigation, N.P.H. and V.S.; data collection, K.S. and K.B.M.; writing—original draft preparation, N.P.H. and K.S.; writing—review and editing, V.S. and N.P.H.; supervision, K.S. All of the authors contributed significantly to the completion of this review, conceiving and designing the study, and writing and improving the paper. All authors have read and agreed to the published version of the manuscript.

**Funding:** This research received no external funding.

**Institutional Review Board Statement:** Not Applicable.

**Informed Consent Statement:** Not Applicable.

**Data Availability Statement:** Not Applicable.

**Acknowledgments:** We would like to express our sincere gratitude to the editor, the anonymous reviewers for their invaluable support and constructive feedback on the manuscript. We also thank the ICFGS Community, a renowned knowledge think-tank on sustainability, for their valuable support throughout the process.

**Conflicts of Interest:** The authors declare no conflict of interest.

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
