# Peer review of "Sustainalism: An Integrated Socio-Economic-Environmental Model to Address Sustainable Development and Sustainability"

_sustainability, doi:10.3390/su151310682_

Round 1

Reviewer 1 Report (Previous Reviewer 2)

Thank you for your efforts in revising the paper. The work remains extremely ambitious and, from my perspective, still requires very significant more work to bring it up to a standard that matches its scope and intentions. Arguments need to be much more thoroughly referenced through, especially with respect to the survey of the various literature.

The discussion of supposed global sustainability principles reads too much like a series of unsubstantiated points.

Unfortunately the work still does not make a significant new contribution to the literature on sustainable development. However, I would encourage the authors to continue.

Further English editing is certainly needed.

Author Response

Please find the attached detailed responses to Reviewer 1 comments  

Reviewer 2 Report (Previous Reviewer 1)

Thank you to the authors for revising the manuscript, which has been improved.

Overall, I think the manuscript can be published as a work that summarises the views on sustainability through the integrated title Sustainalism. 

The authors have changed the theme of the paper, calling their views a new model for addressing sustainability and sustainable development. I would recommend adjusting the title to, for example, "An integrated socio-economic and environmental model to address sustainable development and sustainability", as the components of this model are not individually new, but it is the synthesis of existing views that is important.

Furthermore, the measures proposed by the authors to increase individual responsibility for the sustainable development of society are certainly correct, but in my view are not new. What is needed is a truly new perspective on the mechanisms for achieving sustainable development goals.

Author Response

Please find the attached detailed responses to Reviewer 2 comments 

Round 2

Reviewer 1 Report (Previous Reviewer 2)

Thank you for your further revision of the paper. Unfortunately, my previous comments still stands. The paper is overly ambitious for an article - and while I empathise with the author's intention I see no point in adding yet another term to the sustainability lexicon - sustainalism - when the elements of what is proposed are all well established. Furthermore, much of the paper, and especially toward the end, suggests what should be done but this again covers very familiar ground.

the quality is reasonable but further editing is warranted

Author Response

Dear Reviewer,

Please find the detailed response 

Thank you   

Round 3

Reviewer 1 Report (Previous Reviewer 2)

Thank you for your efforts with the paper. Unfortunately, my perspective on the paper still stands. The paper covers familiar ground - although this could be more approached more systematically than at present.

There is also nothing inherently problematic with the different schools of sustainability thought but the reasons for such differences need to be better addressed. The paper notes:

i. 745ff: Defining Sustainability and sustainable development was a hectic task for the researchers from the beginning. The dispute about the phrase's unclear meaning, since different authors have divergent views on it, has been sparked by the history that links it to "economic growth." Some argue that development, which they define as economic growth, conflicts with sustainability since finite earth cannot support infinite growth[119]. 749
Bringing the holistic approach of sustainable perspectives under a single umbrella is very difficult. The definitions altered gradually throughout the years without losing their soul[105]. Still, the concept of sustainability is waiting for an adequate explanation that will satisfy all sustainability domains.

But why should there be a single definition or approach that satisfies all domains?. In its present form you have a useful bridging concept because it means different things to different people. More particularly, to paraphrase and adapt the saying of Aaron Wildavsky from the policy field if a concept means everything then it means nothing!

As noted previously I do not sufficient merit in the conceptualisation that has occurred or justification for introducing a new term to warrant publication.

some moderate improvements would be helpful at some points for succintness

Author Response

Dear Reviewer, Please find the attached detailed response. Thank you 

This manuscript is a resubmission of an earlier submission. The following is a list of the peer review reports and author responses from that submission.

Round 1

Reviewer 1 Report

I thank my colleagues for the material presented. It is sufficiently systematised, logically structured and interesting to be introduced to sustainability issues, especially for novice researchers and all those interested in the topic.

At the same time, in my opinion, there are questions and comments to the paper.

1. I did not find any novelty. The proposed "Sustainalism" concept has a right to exist, but it contains known approaches and principles for the formation of sustainable development, similar to the views on the construction of communism.

2. The authors initially orient the reader towards broadening the approaches to sustainable development by drawing attention to ethics, culture, etc., but they do not indicate this in principles (8.6), limiting themselves to the known rules of achieving sustainable development.

3. However, the main question I have not found an answer to in the paper is how to change people's mindsets from individualism to collective problem-solving? The authors think of capitalism, socialism and even communism, but forget that individualism is not only in capitalism, but also in socialism. The problem of individualism has not yet been solved in theories about communism either. From these positions the authors' views on the solution of the problem of global sustainability are utopian, as well as the views of many developers of general universal equality.

For these reasons I have decided to reject the manuscript, but I believe it has some value in terms of systematising knowledge and views on the problem of sustainable development. If the emphasis is on this, it is worthwhile to change the title of the work and rework its structure. The introduction of a new theory is still questionable, due to the low arguments for its feasibility.

In any case, thanks again to the authors for returning to the issue of the sustainability of society and the world in general.

P.S. Repeat text: lines 707-710 and 710-712

Reviewer 2 Report

The paper tackles a significant subject. However, after an overview of sustainability writing (which arguably needs to be expanded further given some of the themes of the paper) the article gets to its key points. The problem with sustainalism is that its components are as general as the discussion that has gone before. There is inadequate discussion of why it is different from preceding notions of sustainability. In fact, to me , in some ways this appears as a reinvention of the [sustainability] while, with some elements revisiting much of the writings of the late 60s and early 70s surrounding limits to growth and Schumacher's 'small is beautiful'. However, the fundamental problem is that the arguments are so general - with the sentences appearing often in almost point form - that the argument is insufficient for the claims that are made